# Exchanging Lessons Between Algorithmic Fairness and Domain Generalization

## Abstract

Standard learning approaches are designed to perform well on average for the data distribution available at training time. Developing learning approaches that are not overly sensitive to the training distribution is central to research on domain- or out-of-distribution generalization, robust optimization and fairness. In this work we focus on links between research on domain generalization and algorithmic fairness—where performance under a distinct but related test distributions is studied—and show how the two fields can be mutually beneficial. While domain generalization methods typically rely on knowledge of disjoint "domains" or "environments", "sensitive" label information indicating which demographic groups are at risk of discrimination is often used in the fairness literature. Drawing inspiration from recent fairness approaches that improve worst-case performance without knowledge of sensitive groups, we propose a novel domain generalization method that handles the more realistic scenario where environment partitions are not provided. We then show theoretically and empirically how different partitioning schemes can lead to *increased* or *decreased* generalization performance, enabling us to outperform Invariant Risk Minimization with handcrafted environments in multiple cases. We also show how a re-interpretation of IRMv1 allows us for the first time to directly optimize a common fairness criterion, group-sufficiency, and thereby improve performance on a fair prediction task.

## 1 Introduction

Machine learning achieves super-human performance on many tasks when the test data is drawn from the same distribution as the training data. However, when the two distributions differ, model performance can severely degrade to even below chance predictions (Geirhos et al., 2020). Tiny perturbations can derail classifiers, as shown by adversarial examples (Szegedy et al., 2014) and common-corruptions in image classification (Hendrycks & Dietterich, 2019). Even new test sets collected from the same data acquisition pipeline induce distribution shifts that significantly harm performance (Recht et al., 2019; Engstrom et al., 2020). Many approaches have been proposed to overcome model brittleness in the face of input distribution changes. Robust optimization aims to achieve good performance on any distribution close to the training distribution (Goodfellow et al., 2015; Duchi et al., 2016; Madry et al., 2018). Domain generalization on the other hand tries to go one step further, to generalize to distributions potentially far away from the training distribution.

The field of algorithmic fairness meanwhile primarily focuses on developing metrics to track and mitigate performance differences between different sub-populations or across similar individuals (Dwork et al., 2012; Corbett-Davies & Goel, 2018; Chouldechova & Roth, 2018). Like domain generalization, evaluation using data related to but distinct from the training set is needed to characterize model failure. These evaluations are curated through the design of *audits*, which play a central role in revealing unfair algorithmic decision making (Buolamwini & Gebru, 2018; Obermeyer et al., 2019).

While the ultimate goals of domain generalization and algorithmic fairness are closely aligned, little research has focused on their similarities and how they can inform each other constructively. One of their main common goals can be characterized as:

*Learning algorithms robust to changes across domains or population groups.*

| Method | Handcrafted Environments | Train accs | Test accs |
|---|---|---|---|
| ERM | ✗ | 86.3 ± 0.1 | 13.8 ± 0.6 |
| IRM | ✓ | 71.1 ± 0.8 | 65.5 ± 2.3 |
| **EIIL+IRM** | ✗ | 73.7 ± 0.5 | **68.4 ± 2.7** |

Table 1: Results on CMNIST, a digit classification task where color is a spurious feature correlated with the label during training but anti-correlated at test time. Our method **Environment Inference for Invariant Learning (EIIL)**, taking inspiration from recent themes in the fairness literature, augments IRM to improve test set performance without knowledge of pre-specified environment labels, by instead finding worst-case environments using aggregated data and a reference classifier.

Achieving this not only allows models to generalize to different and unobserved but related distributions, it also mitigates unequal treatment of individuals solely based on group membership.

In this work we explore independently developed concepts from the domain generalization and fairness literatures and exchange lessons between them to motivate new methodology for both fields. Inspired by fairness approaches for unknown group memberships (Kim et al., 2019; Hébert-Johnson et al., 2018; Lahoti et al., 2020), we develop a new domain generalization method that does not require domain identifiers and yet can outperform manual specification of domains (Table 1). Leveraging domain generalization insights in a fairness context, we show the regularizer from IRMv1 (Arjovsky et al., 2019) optimizes a fairness criterion termed "group-sufficiency" which for the first time enables us to explicitly optimize this criterion for non-convex losses in fair classification.

The following contributions show how lessons can be exchanged from the two fields:

- We draw several connections between the goals of domain generalization and those of algorithmic fairness, suggesting fruitful research directions in both fields (Section 2).

- Drawing inspiration from recent methods on inferring worst-case sensitive groups from data, we propose a novel domain generalization algorithm—Environment Inference for Invariant Learning (EIIL)—for cases where training data does not include environment partition labels (Section 3). Our method outperforms IRM on the domain generalization benchmark ColorMNIST without access to environment labels (Section 4).

- We also show that IRM, originally developed for domain generalization tasks, affords a differentiable regularizer for the fairness notion of group sufficiency, which was previously hard to optimize for non-convex losses. On a variant of the UCI Adult dataset where confounding bias is introduced, we leverage this insight with our method EIIL to improve group sufficiency without knowledge of sensitive groups, ultimately improving generalization performance for large distribution shifts compared with a baseline robust optimization method (Section 4).

- We characterize both theoretically and empirically the limitations of our proposed method, concluding that while EIIL can correct a baseline ERM solution that uses a spurious feature or "shortcut" for prediction, it is not suitable for all settings (Sections 3 and 4).

## 2 DOMAIN GENERALIZATION AND ALGORITHMIC FAIRNESS

Here we lay out some connections between the two fields. Table 2 provides a high-level comparison of the objectives and assumptions of several relevant methods. Loosely speaking, recent approaches from both areas share the goal of matching some chosen statistic across a conditioning variable $e$, representing sensitive group membership in algorithmic fairness or an environment/domain indicator in domain generalization. The statistic in question informs the *learning objective* for the resulting model, and is motivated differently in each case. In domain generalization, learning is informed by the properties of the test distribution where good generalization should be achieved. In algorithmic fairness the choice of statistic is motivated by a context-specific *fairness notion*, that likewise encourages a particular solution that achieves "fair" outcomes (Chouldechova & Roth, 2018). Empty spaces in Table 2 suggest areas for future work, and bold-faced entries suggest connections we show in this paper.

| Statistic to match $\forall\, e$ | $e$ known? | Dom-Gen method | Fairness method |
|---|---|---|---|
| $\mathbb{E}[\ell\|e]$ | yes | REx (Krueger et al., 2020), Group DRO (Sagawa et al., 2019) | CVaR Fairness (Williamson & Menon, 2019) |
| $\mathbb{E}[\ell\|e]$ | no | DRO Duchi et al. (2016) | Fairness without Demographics (Hashimoto et al., 2018; Lahoti et al., 2020) |
| $\mathbb{E}[y\|S(x),e]$ | yes | **Score-based IRM** | Group sufficiency (Chouldechova, 2017; Liu et al., 2019) |
| $\mathbb{E}[y\|\Phi(x),e]$ | yes | IRM (Arjovsky et al., 2019) | **IRM across sensitive groups** |
| $\mathbb{E}[y\|\Phi(x),e]$ | no | **EIIL (ours)** | **EIIL (ours)** |
| $\left\|\mathbb{E}[y\|S(x),e]-\mathbb{E}[\hat{y}(x)\|S(x),e]\right\|$ | no | | Multicalibration (Hébert-Johnson et al., 2018) |
| $\left\|\mathbb{E}[y\|e]-\mathbb{E}[\hat{y}(x)\|e]\right\|$ | no | | Multiaccuracy (Kim et al., 2019) |
| $\left\|\mathbb{E}[y\neq\hat{y}(x)\|y=1,e]\right\|$ | no | | Fairness gerrymandering (Kearns et al., 2018) |

Table 2: Domain Generalization (Dom-Gen) and Fairness methods can be understood as matching some statistic across conditioning variable $e$, representing "environment" or "domains" in Dom-Gen literature and "sensitive" group membership in the Fairness literature. We **boldface** new connections highlighted in this work, with blank spaces suggesting future work.

**Notation**   Let $\mathcal{X}$ be the input space, $\mathcal{E}$ the set of environments (a.k.a. "domains"), $\mathcal{Y}$ the target space. Let $x, y, e \sim p^{obs}(x, y, e)$ be observational data, with $x \in \mathcal{X}$, $y \in \mathcal{Y}$, and $e \in \mathcal{E}$. $\mathcal{H}$ denotes a representation space, from which a classifier $w \circ \Phi$ (that maps to the pre-image of $\Delta(\mathcal{Y})$ via a linear map $w$) can be applied. $\Phi : \mathcal{X} \to \mathcal{H}$ denotes the parameterized mapping or "model" that we optimize. We refer to $\Phi(x) \in \mathcal{H}$ as the "representation" of example $x$. $\hat{y} \in \mathcal{Y}$ denotes a hard prediction derived from the classifier by stochastic sampling or probability thresholding. $\ell : \mathcal{H} \times \mathcal{Y} \to \mathbb{R}$ denotes the scalar loss, which guides the learning.

The empirical risk minimization (ERM) solution is found by minimizing the global risk, expressed as the expected loss over the observational distribution:

$$C^{ERM}(\Phi) = \mathbb{E}_{p^{obs}(x,y,e)}[\ell(\Phi(x), y)]. \tag{1}$$

**Domain Generalization**   Domain generalization is concerned with achieving low error rates on unseen test distributions. One way to achieve domain generalization is by casting it as a robust optimization problem (Ben-Tal et al., 2009) where one aims to minimize the worst-case loss for every subset of the training set, or other well-defined perturbation sets around the data (Duchi et al., 2016; Madry et al., 2018). Other approaches tackle domain generalization by adversarially learning representations invariant (Zhang et al., 2017; Hoffman et al., 2018; Ganin et al., 2016) or conditionally invariant (Li et al., 2018) to the environment.

Distributionally Robust Optimization (DRO) (Duchi et al., 2016), seeks good performance for all nearby distributions by minimizing the worst-case loss $\sup_q \mathbb{E}_q[\ell]$ s.t. $D(q||p) < \epsilon$, where $D$ denotes similarity between two distributions (e.g. $\chi^2$ divergence) and $\epsilon$ is a hyperparameter. The objective can be computed as an expectation over $p$ via per-example importance weights $\gamma_i = \frac{q(x_i, y_i)}{p(x_i, y_i)}$.

Recently, *Invariant Learning* approaches such as Invariant Risk Minimization (IRM) (Arjovsky et al., 2019) and Risk Extrapolation (REx) (Krueger et al., 2020) were proposed to overcome the limitations of domain invariant representation learning (Zhao et al., 2019) by discovering invariant relationships between inputs and targets across domains. Invariance serves as a proxy for causality, as features representing "causes" of target labels rather than effects will generalize well under intervention. In IRM, a representation $\Phi(x)$ is learned that performs optimally within each environment—and is thus invariant to the choice of environment $e \in \mathcal{E}$—with the ultimate goal of generalizing to an unknown test dataset $p(x, y|e_{test})$. Because optimal classifiers under standard loss functions can be realized via a conditional label distribution ($f^*(x) = \mathbb{E}[y|x]$), then an invariant representation $\Phi(x)$ must satisfy the following *Environment Invariance Constraint*:

$$\mathbb{E}[y|\Phi(x) = h, e_1] = \mathbb{E}[y|\Phi(x) = h, e_2] \quad \forall\, h \in \mathcal{H} \ \ \forall\, e_1, e_2 \in \mathcal{E}. \qquad \text{(EI-CONSTR)}$$

Intuitively, the representation $\Phi(x)$ encodes features of the input $x$ that induce the same conditional distribution over labels across each environment.

Because trivial representations such as mapping all $x$ onto the same value may satisfy environment invariance, other objectives must be introduced to encourage the predictive utility of $\Phi$. Arjovsky et al. (2019) propose IRM as a way to satisfy (EI-CONSTR) while achieving a good overall risk. As a practical instantiation, the authors introduce IRMv1, a gradient-penalty regularized objective enforcing simultaneous optimality of the same classifier $w \circ \Phi$ in all environments.[1] Denoting by $R^e = \mathbb{E}_{p^{obs}(x,y|e)}[\ell]$ the per-environment risk, the objective to be minimized is

$$C^{IRM}(\Phi) = \sum_{e \in \mathcal{E}} R^e(\Phi) + \lambda ||\nabla_{w|w=1.0} R^e(w \circ \Phi)||^2. \tag{2}$$

Krueger et al. (2020) propose the related Risk Extrapolation (REx) principle, which dictates a stronger preference to exactly equalize $R^e \; \forall \; e$ (e.g. by penalizing variance across $e$), which is shown to improve generalization in several settings.[2]

**Fairness** Early approaches to learning fair representations (Zemel et al., 2013; Edwards & Storkey, 2015; Louizos et al., 2015; Zhang et al., 2018; Madras et al., 2018) leveraged statistical independence regularizers from domain adaptation (Ben-David et al., 2010; Ganin et al., 2016), noting that marginal or conditional independence from domain to prediction relates to the fairness notions of demographic parity $\hat{y} \perp e$ (Dwork et al., 2012) and equal opportunity $\hat{y} \perp e|y$ (Hardt et al., 2016).

Recall that (EI-CONSTR) involves an environment-specific conditional label expectation given a data representation $\mathbb{E}[y|\Phi(x) = h, e]$. Objects of this type have been closely studied in the fair machine learning literature, where $e$ now denotes a "sensitive" indicating membership in a protected demographic group (age, race, gender, etc.), and the vector representation $\Phi(x)$ is typically replaced by a scalar score $S(x) \in \mathbb{R}$. $\mathbb{E}[y|S(x), e]$ can now be interpreted as a *calibration curve* that must be regulated according to some fairness constraint. Chouldechova (2017) showed that equalizing this calibration curve across groups is often incompatible with a common fairness constraint, demographic parity, while Liu et al. (2019) studied "group sufficiency''—satisfied when $\mathbb{E}[y|S(x), e] = \mathbb{E}[y|S(x), e']\forall e, e'$—of classifiers with strongly convex losses, concluding that ERM naturally finds group sufficient solutions without fairness constraints.

Because Liu et al. (2019) consider convex losses, their theoretical results do not hold for neural network representations. However, by noting the link between group sufficiency and the constraint from (EI-CONSTR), we observe that the IRMv1 regularizer (applicable to neural nets) in fact minimizes the group sufficiency gap in the case of a scalar representation $\Phi(x) \subseteq \mathbb{R}$, and when $e$ indicates sensitive group membership. It is worth noting that Arjovsky et al. (2019) briefly discuss using groups as environments, but without specifying a particular fairness criterion.

Reliance on sensitive demographic information is cumbersome since it often cannot be collected without legal or ethical repercussions. Hébert-Johnson et al. (2018) discussed the problem of mitigating subgroup unfairness when group labels are unknown, and proposed *Multicalibration* as a way of ensuring a classifier's calibration curve is invariant to efficiently computable environment splits. Since the proposed algorithm requires brute force enumeration over all possible environments/groups, Kim et al. (2019) suggested a more practical algorithm by relaxing the calibration constraint to an accuracy constraint, yielding a *Multiaccurate* classifier.[3] The goal here is to boost the predictions of a pre-trained classifier through multiple rounds of auditing (searching for worst-case subgroups using an auxiliary model) rather than learning an invariant representation.

A related line of work also leverages inferred subgroup information to improve worst-case model performance using the framework of DRO. Hashimoto et al. (2018) applied DRO to encourage long-term fairness in a dynamical setting where the average loss for a subpopulation influences their propensity to continue engaging with the model. Lahoti et al. (2020) proposed Adversarially Reweighted Learning, which extends DRO using an auxiliary model to compute the importance weights $\gamma_i$ mentioned above. Amortizing this computation mitigates the tendency of DRO to overfit its reweighting strategy to noisy outliers. Wang et al. (2020) proposed a group DRO method for adaptively estimating soft assignments to groups suitable for the setting where group labels are noisy.

---

[1] $w \circ \Phi$ yields a classification decision via linear weighting on the representation features.

[2] Analogous to REx, Williamson & Menon (2019) adapt Conditional Variance at Risk (CVaR) (Rockafellar & Uryasev, 2002) to equalize risk across demographic groups.

[3] Kearns et al. (2018) separately utilize boosting to equalize subgroup errors without sensitive attributes.

**Limitations of generalization-first fairness** One exciting direction for future work is to apply methods developed in the domain generalization literature to tasks where distribution shift is related to some societal harm that should be mitigated. However, researchers should be wary of blind "solutionism", which can be ineffectual or harmful when the societal context surrounding the machine learning system is ignored (Selbst et al., 2019). Moreover, many aspects of algorithmic discrimination are not simply a matter of achieving few errors on unseen distributions. Unfairness due to task definition or dataset collection, as discussed in the study of target variable selection by Obermeyer et al. (2019), may not be reversible by novel algorithmic developments.

## 3 INVARIANCE WITHOUT DEMOGRAPHICS OR ENVIRONMENTS

In this section we draw inspiration from recent work on fair prediction without sensitive labels (Kearns et al., 2018; Hébert-Johnson et al., 2018; Hashimoto et al., 2018; Lahoti et al., 2020) to propose a novel domain generalization algorithm that does not require a priori domain/environment knowledge. To motivate the study of this setting and show the fairness and invariance considerations at play, consider the task of using a high dimensional medical image $x$ to predict a target label $y \in \{0, 1\}$ indicating the presence of COVID-19 in the imaged patient. DeGrave et al. (2020) describe the common use of a composite dataset for this task, where the process of aggregating data across two source hospitals $e \in \{H_1, H_2\}$ leads to a brittle neural net classifier $w \circ \Phi(x)$ that fixates on spurious low-level artifacts in $x$ as predictive features.

Now we will consider a slightly different scenario. Consider a single hospital serving two different demographic populations $e \in \{P_1, P_2\}$. While $P_1$ has mostly sick patients at time $t = 0$ due to the prevalence of COVID-19 in this subpopulation, $P_2$ currently has mostly well patients. Then $p(x, y | e = P_1, t = 0)$ and $p(x, y | e = P_2, t = 0)$ will differ considerably, and moreover a classifier using a spurious feature indicative of subpopulation membership—either a low-level image artifact or attribute of the medical record—may achieve low average error on the available data. Of course such a classifier may generalize poorly. Consider temporal distribution shifts: suppose at time $t = 1$, due to the geographic density of the virus changing over time, $P_1$ has mostly well patients while patients from $P_2$ are now mostly sick. Now the spurious classifier may suffer worse-than-chance error rates and imply unfair outcomes for disadvantaged groups. In reality the early onset and frequency of exposure to COVID-19 has been unequally distributed along many social dimensions (class, race, occupation, etc.) that could constitute protected groups (Tai et al., 2020), raising concerns of additional *algorithmic* discrimination.

Learning to be invariant to spurious features encoding demographics would prevent errors due to such temporal shifts. While loss reweighting as in DRO/ARL can upweight error cases, without an explicit invariance regularizer the model may still do best on average by making use of the spurious feature. IRM can remove the spurious feature in this particular case, but a method for discovering environment partitions directly may occasionally be needed.[4] This need is clear when demographic makeup is not directly observed and a method to sort each example into the maximally separating the spurious feature, i.e. inferring populations $\{P_1, P_2\}$, is needed for effective invariant learning.

### 3.1 ENVIRONMENT INFERENCE FOR INVARIANT LEARNING

We now derive a principle for inferring environments from observational data. Our exposition extends IRMv1 (Equation 2), but we emphasize that our method EIIL is applicable more broadly to any environment-based learning objective. We begin by introducing $\mathbf{u}_i(e') = p^{obs}(e' | x_i, y_i) = \mathbb{1}(e_i = e')$ as an indicator of the hand-crafted environment assignment per-example. Noting that $N_e := \sum_i \mathbf{u}_i(e)$ represents the number of examples in environment $e$, we can re-express this objective to make its dependence on environment labels explicit

$$C^{IRM}(\Phi, \mathbf{u}) = \sum_{e \in \mathcal{E}} \frac{1}{N_e} \sum_i \mathbf{u}_i(e) \ell(\Phi(x_i), y_i) + \sum_{e \in \mathcal{E}} \lambda \left|\left| \nabla_{w|w=1.0} \frac{1}{N_e} \sum_i \mathbf{u}_i(e) \ell(w \circ \Phi(x_i), y_i) \right|\right|_2.$$

(3)

---

[4]In a variant of the first example where hospitals $e \in \{H_1, H_2\}$ are known, the given environments could be *improved* by a method that sorts whether spurious artifacts are present, i.e. inferring equipment type.

Our general strategy is to replace the binary indicator $\mathbf{u}_i(e)$, with a probability distribution $q(e|x_i, y_i)$, representing a soft assignment of the $i$-th example to the $e$-th environment. $q(e|x_i, y_i)$ should capture worst-case environments w.r.t the invariant learning objective; rewriting $q(e|x_i, y_i)$ as $\mathbf{q}_i(e)$ for consistency with the above expression, we arrive at the following bi-level optimization:

$$\min_\Phi \max_\mathbf{q} C^{IRM}(\Phi, \mathbf{q}). \qquad \text{(EIIL)}$$

We leave the full exploration of this bi-level optimization to future work, but for now propose the following practical sequential approach, which we call **EIILv1** (See Appendix A for pseudocode):

1. Input *reference model* $\tilde{\Phi}$;

2. Fix $\Phi \leftarrow \tilde{\Phi}$ and fully optimize the inner loop of (EIIL) to infer environments $\tilde{\mathbf{q}}_i(e) = \tilde{q}(e|x_i, y_i)$;

3. Fix $\mathbf{q} \leftarrow \tilde{\mathbf{q}}$ and fully optimize the outer loop to yield the new model $\Phi$.

Instead of requiring hand-crafted environments, we instead require a trained reference model $\tilde{\Phi}$, which is arguably easier to produce and could be found using ERM on $p^{obs}(x, y)$, for example. In our experiments we consider binary environments[5] and explicitly parameterize the $q(e|x, y)$ as a vector of probabilities for each example in the training data.[6]

## 3.2 ANALYZING THE EIIL SOLUTION

To characterize the ability of EIILv1 to generalize to unseen test data, we now examine the inductive bias for generalization provided by the reference model $\tilde{\Phi}$. We state the main result here and defer the proofs to Appendix B. Consider a dataset with some feature(s) $z$ which are spurious, and other(s) $v$ which are valuable/causal w.r.t. the label $y$. Our proof considers binary features/labels and two environments, but the same argument extends to other cases. Our goal is to find a model $\Phi$ whose representation $\Phi(v, z)$ is invariant w.r.t. $z$ and focuses solely on $v$.

**Theorem 1** *Consider environments that differ in the degree to which the label $y$ agrees with the spurious features $z$: $\mathbb{P}(\mathbb{1}(y = z)|e_1) \neq \mathbb{P}(\mathbb{1}(y = z)|e_2)$: then a reference model $\tilde{\Phi}_{Spurious}$ that is invariant to valuable features $v$ and solely focusing on spurious features $z$ maximally violates the Invariance Principle (EI-CONSTR). Likewise, consider the case with fixed representation $\Phi$ that focuses on the spurious features: then a choice of environments that maximally violates (EI-CONSTR) is $e_1 = \{v, z, y|\mathbb{1}(y = z)\}$ and $e_2 = \{v, z, y|\mathbb{1}(y \neq z)\}$.*

If environments are split according to agreement of $y$ and $z$, then the constraint from (EI-CONSTR) is satisfied by a representation that ignores $z$: $\Phi(x) \perp z$. Unfortunately this requires a priori knowledge of either the spurious feature $z$ or a reference model $\tilde{\Phi}_{Spurious}$ that extracts it. When the *wrong* solution $\tilde{\Phi}_{Spurious}$ is not a priori known, it can sometimes be recovered directly from the training data; for example in CMNIST we find that $\tilde{\Phi}_{ERM}$ approximates $\tilde{\Phi}_{Color}$. This allows EIIL to find environment partitions providing the starkest possible contrast for invariant learning.

Even if environment partitions are available, it may be possible to improve performance by inferring new partitions from scratch. It can be shown (see Appendix B.2) that the environments provided in the CMNIST dataset (Arjovsky et al., 2019) do not maximally violate (EI-CONSTR) for a reference model $\tilde{\Phi}_{Color}$, and are thus not maximally informative for learning to ignore color. Accordingly, EIIL improves test accuracy for IRM compared with the hand-crafted environments (Table 1).

---

[5]The theoretical analysis of IRM suggests that the more (statistically independent) environments the better in term of generalization guarantees. This suggests in the setting where these analyses apply, extending EIIL to find more than two environments (with a term to promote diversity amongst inferred environments) may further help out-of-domain generalization, which we leave for future investigation.

[6]Note that under this parameterization, when optimizing the inner loop with fixed $\Phi$ the number of parameters equals the number of data points (which is small relative to standard neural net training). We leave amortization of $q$ to future work.

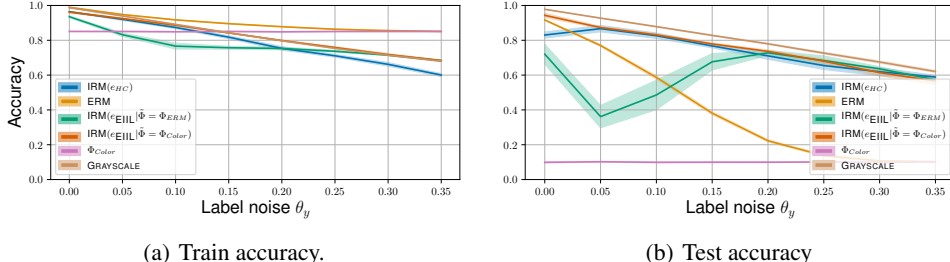

(a) Train accuracy.

(b) Test accuracy

Figure 1: CMNIST with varying label noise $\theta_y$. Under high label noise ($\theta_y > .2$), where the spurious feature color correlates to label *more* than shape on the train data, IRM($e_{\text{EIIL}}$) matches or exceeds the performance of IRM($e_{\text{HC}}$) on the test set *without relying on hand-crafted environments*. Under medium label noise ($.1 < \theta_y < .2$), IRM($e_{\text{EIIL}}$) is worse than IRM($e_{\text{HC}}$) but better than ERM, the logical approach if environments are not available. Under low label noise ($\theta_y < .1$), where color is *less* predictive than shape at train time, ERM performs well and IRM($e_{\text{EIIL}}$) fails. GRAYSCALE indicates the oracle solution using shape alone, while $\Phi_{Color}$ uses color alone.

## 4 EXPERIMENTS

We defer a proof-of-concept synthetic regression experiment to Appendix E for lack of space. We proceed with the established domain generalization benchmark ColorMNIST, and then discuss a variant of the algorithmic fairness dataset UCI Adult. We note that benchmarking model performance on a shifted test distribution without access to validation samples—especially during model selection—is a difficult open problem, a solution to which is beyond the scope of this paper. Accordingly we use the default IRM hyperparameters wherever appropriate, and otherwise follow a recently proposed model selection strategy (Gulrajani & Lopez-Paz, 2020) (see Appendix D).[7]

### 4.1 COLORMNIST

ColorMNIST (CMNIST) is a noisy digit recognition task[8] where color is a spurious feature that correlates with the label at train time but anti-correlates at test time, with the correlation strength at train time varying across two pre-specified environments (Arjovsky et al., 2019). Crucially, label noise is applied by flipping $y$ with probability $\theta_y$; the default setting ($\theta_y = 0.25$) implies that shape (the correct feature) is marginally less reliable than color in the train set, so naive ERM ignores shape to focus on color and suffers from below-chance performance at test time.

We evaluated the performance of the following methods: **ERM:** A naive MLP that does not make use of environment labels $e$, but instead optimizes the average loss on the aggregated environments; **IRM($e_{\text{HC}}$):** the method of Arjovsky et al. (2019) using hand-crafted environment labels; **IRM($e_{\text{EIIL}}$):** our proposed method (a.k.a. EIILv1) that infers useful environments (not using hand-crafted environment labels) based on the naive ERM, then applies IRM to the inferred environments.

After noting that EIILv1—denoted IRM($e_{\text{EIIL}}$) above—outperforms IRM *without access to environment labels* in the default setting (See Tables 1 and 6), we examine how the various methods perform as a function of $\theta_y$. This parameter influences the ERM solution since low $\theta_y$ implies shape is more reliable than color in the aggregated training data (thus ERM generalizes well), while the opposite trend holds for high $\theta_y$. Because EIILv1 relies on a reference model $\tilde{\Phi}$, its performance is also affected when $\tilde{\Phi} = \text{ERM}$ (Figure 1). We find that IRM($e_{\text{EIIL}}$) generalizes better than IRM($e_{\text{HC}}$) with sufficiently high label noise $\theta_y > .2$, but generalizes poorly under low label noise. This is precisely due to the success of ERM in this setting, where shape is a more reliable feature in the training data than color. We verify this conclusion by evaluating IRM($e_{\text{EIIL}}$) when $\tilde{\Phi} = \Phi_{Color}$, i.e. a hand-coded color-based predictor as reference. This does relatively well across all settings of $\theta_y$, approaching the performance of the (oracle) baseline that classifies using grayscale inputs.

---

[7]Following the suggestion of Gulrajani & Lopez-Paz (2020), we note that Section 4.2 contains "oracle" results that are overly optimistic for each method (see Appendix D for model selection details).

[8]MNIST digits are grouped into $\{0, 1, 2, 3, 4\}$ and $\{5, 6, 7, 8, 9\}$ so the CMNIST target label $y$ is binary.

## 4.2 CENSUS DATA

We now study a fair prediction problem using a variant of the UCI Adult dataset,[9] which comprises $48,842$ individual census records collected from the United States in 1994. The task commonly used as an algorithmic fairness benchmark is to predict a binarized income indicator (thresholded at $\$50,000$) as the target label, possibly considering sensitive attributes such as age, sex, and race. Because the original task measures in-distribution test performance, we instead construct a variant of this dataset suitable for measuring out-of-distribution test performance, which we call ConfoundedAdult.

|  | Train accs | Test accs |
|---|---|---|
| Baseline | $92.7 \pm 0.5$ | $31.1 \pm 4.4$ |
| ARL (Lahoti et al., 2020) | $72.1 \pm 3.6$ | $61.3 \pm 1.7$ |
| EIILv1 | $69.7 \pm 1.6$ | $\mathbf{78.8 \pm 1.4}$ |

Table 3: Accuracy on ConfoundedAdult, a variant of the UCI Adult dataset where some sensitive subgroups correlate to the label at train time and reverse this correlation pattern at test time.

Lahoti et al. (2020) demonstrate the benefit of per-example loss reweighting on UCI Adult using their method ARL to improve predictive performance for undersampled subgroups. Following Lahoti et al. (2020), we consider the effect of four sensitive *subgroups*—defined by composing binarized race and sex labels—on model performance, assuming the model does not know a priori which features are sensitive. However, we focus on a distinct generalization problem where a pernicious dataset bias confounds the training data, making subgroup membership predictive of the label on the training data. At test time these correlations are reversed, so a predictor that infers subgroup membership to make predictions will perform poorly at test time (see Appendix C for details). Dwork et al. (2012) described a similar motivating scenario where the conditional distribution mapping features to target labels varies across demographic groups due to cultural differences, so the most predictive predictor for one group may not generalize to the others. The large distribution shift of our test set can be understood as a worst-case *audit* to determine whether the classifier uses subgroup information in its predictions.

Using EIILv1—to first infer worst-case environments then ensure invariance across them—performs favorably on the audit test set, compared with ARL and a baseline MLP (Table 3). We also find that, without access to sensitive group information, using the IRMv1 penalty on the EIIL environments *improves* subgroup sufficiency (Figure 2). Appendix E.3 provides an ablation showing that all components of the EIILv1 approach are needed to achieve the best performance.

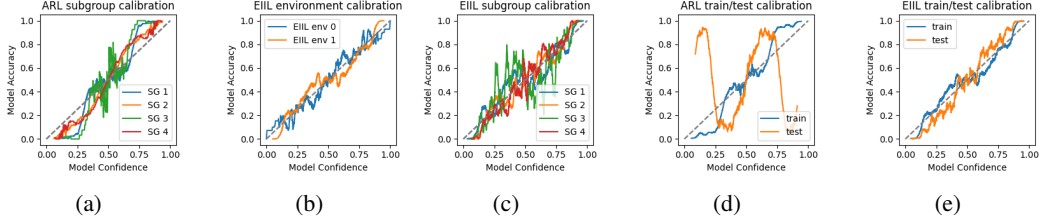

(a)  (b)  (c)  (d)  (e)

Figure 2: We examine *subgroup sufficiency*—whether calibration curves match across demographic subgroups—on the ConfoundedAdult dataset. Whereas ARL is not subgroup-sufficient (a), EIIL infers worst-case environments and regularizes their calibration to be similar (b), ultimately improving subgroup sufficiency (c). This helps EIIL generalize better to a shifted test set (e) compared with ARL (d). Note that neither method uses sensitive group information during learning.

---

[9]https://archive.ics.uci.edu/ml/datasets/adult

## 5 CONCLUSION

We discussed the common goals of algorithmic fairness and domain generalization, compared related methods from each literature, and suggested how lessons can be exchanged between the two fields to inform future research. The most concrete outcome of this discussion was our novel domain generalization method, Environment Inference for Invariant Learning (EIIL). Drawing inspiration from fairness methods that optimize worst-case performance without access to demographic information, EIIL improves the performance of IRM on CMNIST without requiring a priori knowledge of the environments. On a variant of the UCI Adult dataset, EIIL makes use of the IRMv1 regularizer to improve group sufficiency—a fairness criterion previously difficult to optimize for non-convex losses—without requiring knowledge of the sensitive groups.

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

## A EIILv1 PSUEDOCODE

---

**Algorithm 1** The first stage of EIILv1 infers two environments that maximally violate the IRM objective. The inferred environments are then used to train an IRM solution from scratch.

---

**Input:** Reference model $\tilde{\Phi}$, dataset $\mathcal{D} = \{x_i, y_i\}$ with $x_i, y_i \sim p^{obs}$ iid, loss function $\ell$, duration $N_{steps}$

**Output:** Worst case data splits $\mathcal{D}_1, \mathcal{D}_2$ for use with IRM.

Randomly initialize $\mathbf{e} \in \mathbb{R}^{|\mathcal{D}|}$ as vectorized logit of posterior with $\sigma(\mathbf{e}_i) := q(e|x_i, y_i)$. **for** $n \in 1 \ldots N_{steps}$ **do**

$\quad R^1 = \frac{1}{\sum_{i'} \sigma(\mathbf{e}_{i'})} \sum_i \sigma(\mathbf{e}_i) \ell(\tilde{\Phi}(x_i), y_i)$ ;  // D1 risk

$\quad G^1 = \nabla_{\mathbf{w}|\mathbf{w}=1} \|\frac{1}{\sum_{i'} \sigma(\mathbf{e}_{i'})} \sum_i \sigma(\mathbf{e}_i) \ell(\mathbf{w} \circ \tilde{\Phi}(x_i), y_i)\|^2$ ;  // D1 invariance regularizer

$\quad R^2 = \frac{1}{\sum_{i'} 1 - \sigma(\mathbf{e}_{i'})} \sum_i (1 - \sigma(\mathbf{e}_i)) \ell(\tilde{\Phi}(x_i), y_i)$ ;  // D2 risk

$\quad G^2 = \nabla_{\mathbf{w}|\mathbf{w}=1} \|\frac{1}{\sum_{i'} 1 - \sigma(\mathbf{e}_i)} \sum_i (1 - \sigma(\mathbf{e}_i)) \ell(\mathbf{w} \circ \tilde{\Phi}(x_i), y_i)\|^2$ ;  // D2 invariance regularizer

$\quad L = \frac{1}{2} \sum_{e \in \{1,2\}} R^e + \lambda G^e$

$\quad \mathbf{e} \leftarrow OptimUpdate(\mathbf{e}, \nabla_{\mathbf{e}} L)$

**end**

$\hat{\mathbf{e}} \sim Bernoulli(\sigma(\mathbf{e}))$ ;  // sample splits

$\mathcal{D}_1 \leftarrow \{x_i, y_i | \hat{\mathbf{e}}_i = 1\}, \mathcal{D}_2 \leftarrow \{x_i, y_i | \hat{\mathbf{e}}_i = 0\}$ ;  // split data

---

# B    PROOFS

## B.1    PROOF OF THEOREM 1

Consider a dataset with some feature(s) $z$ which are spurious, and other(s) $v$ which are valuable/causal w.r.t. the label $y$. This includes data generated by models where $v \to y \to z$, such that $P(y|v, z) = P(y|v)$. Assume further that the observations $x$ are functions of both spurious and valuable features: $x := f(v, z)$. The aim of invariant learning is to form a classifier that predicts $y$ from $x$ that focuses solely on the causal features, i.e., is invariant to $z$ and focuses solely on $v$.

Consider a classifier that produces a score $S(x)$ for example $x$. In the binary classification setting $S$ is analogous to the model $\Phi$, while the score $S(x)$ is analogous to the representation $\Phi(x)$. To quantify the degree to which the constraint in the Invariant Principle (EI-CONSTR) holds, we introduce a measure called the *group sufficiency gap*[10]:

$$\Delta(S, e) = \mathbb{E}[[\mathbb{E}(y|S(x), e_1) - \mathbb{E}(y|S(x), e_2)]]$$

Now consider the notion of an environment: some setting in which the $x \to y$ relationship varies (based on spurious features). Assume a single binary spurious feature $z$. We restate Theorem 1 as follows:

Claim: If environments are defined based on the agreement of the spurious feature $z$ and the label $y$, then a classifier that predicts based on $z$ alone maximizes the group-sufficiency gap (and vice versa – if a classifier predicts $y$ directly by predicting $z$, then defining two environments based on agreement of label and spurious feature—$e_1 = \{v, z, y | \mathbb{1}(y = z)\}$ and $e_2 = \{v, z, y | \mathbb{1}(y \neq z)\}$—maximizes the gap).

We can show this by first noting that if the environment is based on spurious feature-label agreement, then with $e \in \{0, 1\}$ we have $e = \mathbb{1}(y = z)$. If the classifier predicts $z$, i.e. $S(x) = z$, then we have

$$\Delta(S, e) = \mathbb{E}[\mathbb{E}[y|z(x), \mathbb{1}(y = z)] - \mathbb{E}[y|z(x), \mathbb{1}(y \neq z)]]$$

For each instance of $x$ either $z = 0$ or $z = 1$. Now we note that when $z = 1$ we have $\mathbb{E}(y|z, \mathbb{1}(y = z)) = 1$ and $\mathbb{E}(y|z, \mathbb{1}(y \neq z)) = 0$, while when $z = 0$ $E(y|z, I[y == z]) = 0$ and $\mathbb{E}[y|z, \mathbb{1}(y \neq z)] = 1$. Therefore for each example $|E(y|z(x), \mathbb{1}(y = z)) - E(y|z(x), \mathbb{1}(y \neq z)| = 1$, contributing to an overall $\Delta(S, e) = 1$, which is the maximum value for the sufficiency gap.

## B.2    GIVEN CMNIST ENVIRONMENTS ARE SUBOPTIMAL W.R.T. SUFFICIENCY GAP

The regularizer from IRMv1 encourages a representation for which sufficiency gap is minimized between the available environments. Therefore when faced with a new task it is natural to measure the natural sufficiency gap between these environments, mediated through a naive or baseline method. Here we show that for CMNIST, when considering a naive color-based classifier as the reference model, the given environment splits are actually *suboptimal* w.r.t. sufficiency gap, which motivates the proposed EIIL approach for inferring environments that have a more sever sufficiency gap for the reference model.

We begin by computing $\Delta(S, e)$, the sufficiency gap for color-based classifier $g$ over the given train environments $\{e_1, e_2\}$. We introduce an auxiliary color variable $z$, which is not observed but can be sampled from via the color based classifier $g$:

$$p(y|g(x) = x', e) = \mathbb{E}_{p(z|x')} [p(y|z, e, x').]$$

---

[10]This was previously used in a fairness setting by Liu et al. (2019) to measure differing calibration curves across groups.

Denote by GREEN and RED the set of green and red images, respectively. I.e. we have $z \in G$ iff $z = 1$ and $x \in$ GREEN iff $z(x) = 1$. The the sufficiency gap is expressed as

$$\Delta(S, e) = \mathbb{E}_{p(x,e)} \left[ \left| \mathbb{E}_{p(y|x,e_1)}[y|g(x), e_1] - \mathbb{E}_{p(y|x,e_2)}[y|g(x), e_2] \right| \right]$$

$$= \mathbb{E}_{p(z,e)} \left[ \left| \mathbb{E}_{p(y|z,e_1)}[y|z, e_1] - \mathbb{E}_{p(y|z,e_2)}[y|z, e_2] \right| \right]$$

$$= \frac{1}{2} \sum_{z \in \{\text{GREEN}, \text{RED}\}} \left[ \left| \mathbb{E}_{p(y|z,e_1)}[y|z, e_1] - \mathbb{E}_{p(y|z,e_2)}[y|z, e_2] \right| \right]$$

$$= \frac{1}{2} (|\mathbb{E}[y|z = \text{GREEN}, e_1] - \mathbb{E}[y|z = \text{GREEN}, e_2]| + |\mathbb{E}[y|z = \text{RED}, e_1] - \mathbb{E}[y|z = \text{RED}, e_2]|)$$

$$= \frac{1}{2} (|0.1 - 0.2| - |0.9 - 0.8|)$$

$$= \frac{1}{10}.$$

The regularizer in IRMv1 is trying to reduce the sufficiency gap, so in some sense we can think about this gap as a learning signal for the IRM learner. A natural question would be whether a different set of environment partition $\{e\}$ can be found such that this learning signal is stronger, i.e. the sufficiency gap is increased. We find the answer is yes. Consider an environment distribution $q(e|x, y, z)$ that assigns each data point to one of two environments. Any assignment suffices so far as its marginal matches the observed data: $\int_z \int_e q(x, y, z, e) = p^{\text{obs}}(x, y)$.

We can now express the sufficiency gap (given a color-based classifier $g$) as a function of the environment assignment $q$:

$$\Delta(S, e \sim q) = \mathbb{E}_{q(x,e)} [|\mathbb{E}_{q(y|x,e,x)}[y|g(x), e_1] - \mathbb{E}_{q(y|x,e,x)}[y|g(x), e_2]|]$$

$$= \mathbb{E}_{q(x,e)} [|\mathbb{E}_{q(y|z,e,x)p(z|x)}[y|z, e_1] - \mathbb{E}_{q(y|z,e,x)p(z|x)}[y|z, e_2]|]$$

Where we use the same change of variables trick as above to replace $g(x)$ with samples from $p(z|x)$ (note that this is the color factor from the generative process $p$ according with our assumption that $g$ matches this distribution).

We want to show that there exists a $q$ yielding a higher sufficiency gap than the given environments. Consider $q$ that yields the conditional label distribution

$$q(y|x, e, z) := q(y|e, z) = \begin{cases} \mathbb{1}(y = z) \text{ if } e = e_1, \\ \mathbb{1}(y \neq z) \text{ if } e = e_2. \end{cases}$$

This can be realized by an encoder/auditor $q(e|x, y, z)$ that ignores image features in $x$ and partitions the example based on whether or not the label $y$ and color $z$ agree. We also note that $z$ is deterministically the color of the image in the generative process: $p(z|x) = \mathbb{1}(x = \text{RED})$

Now we can compute the sufficiency gap:

$$
\begin{aligned}
\Delta(S, e \sim q) &= \mathbb{E}_{q(x,e)}[|\mathbb{E}_{q(y|z,e,x)p(z|x)}[y|z,e_1] - \mathbb{E}_{q(y|z,e,x)p(z|x)}[y|z,e_2]|] \\
&= \frac{1}{2}\mathbb{E}_{x \in \text{\textcolor{red}{RED}}}|\mathbb{E}_{q(y|z,e,x)p(z|x)}[y|z,e_1] - \mathbb{E}_{q(y|z,e,x)p(z|x)}[y|z,e_2]| \\
&\quad + \frac{1}{2}\mathbb{E}_{x \in \text{\textcolor{green}{GREEN}}}|\mathbb{E}_{q(y|z,e,x)p(z|x)}[y|z,e_1] - \mathbb{E}_{q(y|z,e,x)p(z|x)}[y|z,e_2]| \\
&= \frac{1}{2}\mathbb{E}_{x \in \text{\textcolor{red}{RED}}}(|\sum_y \sum_z (y * \mathbb{1}(y = z) * \mathbb{1}(g(x) = z)) - \sum_y \sum_z (y * \mathbb{1}(y \neq z) * \mathbb{1}(g(x) = z))|) \\
&\quad + \mathbb{E}_{x \in \text{\textcolor{green}{GREEN}}}\frac{1}{2}(|\sum_y \sum_z (y * \mathbb{1}(y = z) * \mathbb{1}(g(x) = z)) - \sum_y \sum_z (y * \mathbb{1}(y \neq z) * \mathbb{1}(g(x) = z))|) \\
&= \frac{1}{2}\mathbb{E}_{x \in \text{\textcolor{red}{RED}}}(|\sum_y (y * \mathbb{1}(y = 1) * \mathbb{1}(x \in \text{\textcolor{red}{RED}})) - \sum_y (y * \mathbb{1}(y \neq 1) * \mathbb{1}(x \in \text{\textcolor{red}{RED}}))|) \\
&\quad + \mathbb{E}_{x \in \text{\textcolor{green}{GREEN}}}\frac{1}{2}(|\sum_y \sum_z (y * \mathbb{1}(y = 0) * \mathbb{1}(x \in \text{\textcolor{green}{GREEN}})) - \sum_y \sum_z (y * \mathbb{1}(y \neq 0) * \mathbb{1}(x \in \text{\textcolor{green}{GREEN}}))|) \\
&= \frac{1}{2}\mathbb{E}_{x \in \text{\textcolor{red}{RED}}}[|1 - 0|] + \mathbb{E}_{x \in \text{\textcolor{green}{GREEN}}}[\frac{1}{2}|0 - 1|] = \frac{1}{2} + \frac{1}{2} = 1.
\end{aligned}
$$

Note that 1 is the maximal sufficiency gap, meaning that the described environment partition maximizes the sufficiency gap w.r.t. the color-based classifier $g$.

## C   DATASET DETAILS

**Constructing the ConfoundedAdult dataset**   To create our semi-synthetic dataset, called ConfoundedAdult, we start by observing that the conditional distribution over labels varies across the subgroups, and in some cases subgroup membership is very predictive of the target label. We construct a test set (a.k.a. the audit set) where this relationship between subgroups and target label is reversed.

The four sensitive subgroups are defined following the procedure of Lahoti et al. (2020), with sex (recorded as binary: Male/Female) and binarized race (Black/non-Black) attributes compose to make four possible subgroups: Non-Black Males (SG1), Non-Black Females (S2), Black Males (SG3), and Black Females (SG4).

We start with the observation that each subgroup has a different correlation strength with the target label, and in some cases subgroup membership alone can be used to achieve relatively low error rates in prediction. As these correlations should be considered "spurious" to mitigate unequal treatment across groups, we create a semi-synthetic variant of the UCI Adult dataset, which we call ConfoundedAdult, where these spurious correlations are exaggerated. Table 4 shows various conditional label distributions for the original dataset and our proposed variant. The test set for ConfoundedAdult revereses the correlation strengths, which can be thought of as a worst-case audit to ensure the model is not relying on subgroup membership alone in its predictions. We generate samples for ConfoundedAdult using importance sampling, keeping the original train/test splits from UCI Adult as well as the subgroup sizes, but sampling individual examples under/over-sampled according to importance weights $\frac{p^{ConfoundedAdult}}{p^{UCIAdult}}$.

## D   EXPERIMENTAL DETAILS

**Model selection**   Krueger et al. (2020) discussed the pitfalls of achieving good test performance on CMNIST by using test data to tune hyperparameters. Because our primary interest is in the properties of the inferred environment rather than the final test performance, we sidestep this issue in the Synthetic Regression and CMNIST experiments by using the default parameters of IRM without further tuning. However for the ConfoundedAdult dataset a specific strategy for model selection is needed.

| Subgroup ($SG$) | $p(y = 1|SG)$ | | | |
|---|---|---|---|---|
| | UCIAdult | | ConfoundedAdult | |
| | Train | Test | Train | Test |
| 1 | 0.31 | 0.30 | 0.94 | 0.06 |
| 2 | 0.11 | 0.12 | 0.06 | 0.94 |
| 3 | 0.19 | 0.16 | 0.94 | 0.06 |
| 4 | 0.06 | 0.04 | 0.06 | 0.94 |

Table 4: ConfoundedAdult is a variant of the UCI Adult dataset that emphasizes test-time distrbution shift.

We refer the interested reader to Gulrajani & Lopez-Paz (2020) for an extensive discussion of possible model selection strategies. They also provide a large empirical study showing that ERM is difficult baseline to beat when all methods are put on equal footing w.r.t. model selection.

In our case, we use the most relaxed model selection method proposed by Gulrajani & Lopez-Paz (2020), which amounts to allowing each method a 20 test evaluations using hyperparameter chosen at random from a reasonable range, with the best hyperparameter setting selected for each method. While none of the methods is given an unfair advantage in the search over hyperparameters, the basic model selection premise does not translate to real-world applications, since information about the test-time distribution is required to select hyperparameters. Thus these results can be understood as being overly optimistic for each method, although the relative ordering between the methods can still be compared.

**CMNIST** IRM is trained on these two environments and tested on a holdout environment constructed from $10,000$ test images in the same way as the training environments, where colour is predictive of the noisy label 10% of the time. So using color as a feature to predict the label will lead to an accuracy of roughly 10% on the test environment, while it yields 80% and 90% accuracy respectively on the training environments.

To evaluate IRM($e_{\text{EIIL}}$) we remove the environment identifier from the training set and thus have one training set comprised of $50,000$ images from both original training environments. We then train an MLP with binary cross-entropy loss on the training environments, freeze its weights and use the obtained model to learn environment splits that maximally violate the IRM penalty. When optimizing the inner loop of EIIL, we use Adam with learning rate 0.001 for $10,000$ steps with full data batches used to computed gradients.

The obtained environment partitions are then used to train a new model from scratch with IRM. Following Arjovsky et al. (2019), we allow the representation to train for several hundred annealing steps before applying the IRMv1 penalty.

**Census data** Following Lahoti et al. (2020), we use a two-hidden-layer MLP architecture for all methods, with 64 and 32 hidden units respectively, and a linear adversary for ARL. We optimize all methods using Adagrad; learning rates, number of steps, and batch sizes chosen by the model selection strategy described above (with 20 test evaluations per method), as are penalty weights for IRMv1 regularizer and standard weight decay. For the inner loop of EIIL (inferring the environments), we use the same settings as in CMNIST. We find that the performance of EIIL is somewhat sensitive to the number of steps taken with the IRMv1 penalty applied. To limit the number of test queries needed during model selection, we use an early stopping heuristic by enforcing the IRMv1 penalty only during the final 500 steps of training, with the previous steps serving as annealing period to learn a baseline representation to be regularized. Unlike the previous datasets, here we use minibatches to compute gradients during IRM training (for consistency with the ARL method, which uses minibatches). However, full batch gradients are still used for inferring environments in EIIL.

|  | Causal MSE | Noncausal MSE |
|---|---|---|
| ERM | $0.827 \pm 0.185$ | $0.824 \pm 0.013$ |
| ICP($e_{\text{HC}}$) | $1.000 \pm 0.000$ | $0.756 \pm 0.378$ |
| IRM($e_{\text{HC}}$) | $0.666 \pm 0.073$ | $0.644 \pm 0.061$ |
| **IRM($e_{\text{EIIL}}$)** | $\mathbf{0.148 \pm 0.185}$ | $\mathbf{0.145 \pm 0.177}$ |

Table 5: IRM using EIIL-discovered environments ($e_{\text{EIIL}}$) outperforms IRM in a synthetic regression setting without the need for hand-crafted environments ($e_{\text{HC}}$). This is because the reference representation $\tilde{\Phi} = \Phi_{\text{ERM}}$ uses the spurious feature for prediction. MSE + standard deviation across 5 runs reported.

# E  ADDITIONAL EMPIRICAL RESULTS

## E.1  SYNTHETIC DATA

We begin with a regression setting originally used as a toy dataset for evaluating IRM (Arjovsky et al., 2019). The features $\mathbf{x} \in \mathbb{R}^N$ comprise a "causal" feature $\mathbf{v} \in \mathbb{R}^{N/2}$ concatenated with a "non-causal" feature $\mathbf{z} \in \mathbb{R}^{N/2}$: $\mathbf{x} = [\mathbf{v}, \mathbf{z}]$. Noise varies across hand-crafted environments $e$:

$$\mathbf{v} = \epsilon_{\mathbf{v}} \qquad\qquad \epsilon_{\mathbf{v}} \sim \mathcal{N}(0, 25)$$
$$\mathbf{y} = \mathbf{v} + \epsilon_{\mathbf{y}} \qquad\qquad \epsilon_{\mathbf{y}} \sim \mathcal{N}(0, e^2)$$
$$\mathbf{z} = \mathbf{y} + \epsilon_{\mathbf{z}} \qquad\qquad \epsilon_{\mathbf{z}} \sim \mathcal{N}(0, 1).$$

We evaluated the performance of the following methods:

- **ERM:** A naive regressor that does not make use of environment labels $e$, but instead optimizes the average loss on the aggregated environments;

- **IRM($e_{\text{HC}}$):** the method of Arjovsky et al. (2019) using hand-crafted environment labels;

- **ICP($e_{\text{HC}}$):** the method of Peters et al. (2016) using hand-crafted environment labels;

- **IRM($e_{\text{EIIL}}$):** our proposed method (which does use hand-crafted environment labels) that infers useful environments based on the naive ERM, then applies IRM to the inferred environments.

The regression methods fit a scalar target $y = \mathbf{1}^T \mathbf{y}$ via a regression model $\hat{y} \approx \mathbf{w}^T \mathbf{x}$ to minimize $||y - \hat{y}||$ w.r.t. $\mathbf{w}$, plus an invariance penalty as needed. The optimal (causally correct) solution is $\mathbf{w}^* = [\mathbf{1}, \mathbf{0}]$ Given a solution $[\hat{\mathbf{w}}_v, \hat{\mathbf{w}}_z]$ from one of the methods, we report the mean squared error for the causal and non-causal dimensions as $||\hat{\mathbf{w}}_v - \mathbf{1}||_2^2$ and $||\hat{\mathbf{w}}_z - \mathbf{0}||_2^2$ (Table 5). Because $\mathbf{v}$ is marginally noisier than $\mathbf{z}$, ERM focuses on the spurious $\mathbf{z}$. IRM using hand-crafted environments, denoted IRM($e_{\text{HC}}$), exploits variability in noise level in the non-causal feature (which depends on the variability of $\sigma_{\mathbf{y}}$) to achieve lower error. Using EIILv1 instead of hand crafted environments yields an improvement on the resulting IRM solution (denoted IRM($e_{\text{EIIL}}$)) by learning worst-case environments for invariant training.

We show in a follow-up experiment that the EIILv1 solution is indeed sensitive to the choice of reference representation, and in fact, can only discover useful environments (environments that allow IRM($e_{\text{EIIL}}$)to learn the correct causal representation) when the reference representation encodes the *incorrect* inductive bias by focusing on the spurious feature. We can explore this dependence of EIILv1 on the mix of spurious and non-spurious features in the reference model by constructing a $\tilde{\Phi}$ that varies in the degree it focuses on the spurious feature, according to convex mixing parameter $\alpha \in [0, 1]$. $\alpha = 0$ indicates focusing entirely on the correct causal feature , while $\alpha = 1$ indicates focusing on the spurious feature . We refer to this variant as IRM($e_{\text{EIIL}}|\tilde{\Phi} = \Phi_{\alpha-\text{SPURIOUS}}$), and measure its performance as a function of $\alpha$ (Figure 3). Environment inference only yields good test-time performance for high values of $\alpha$, where the reference model captures the *incorrect* inductive bias.

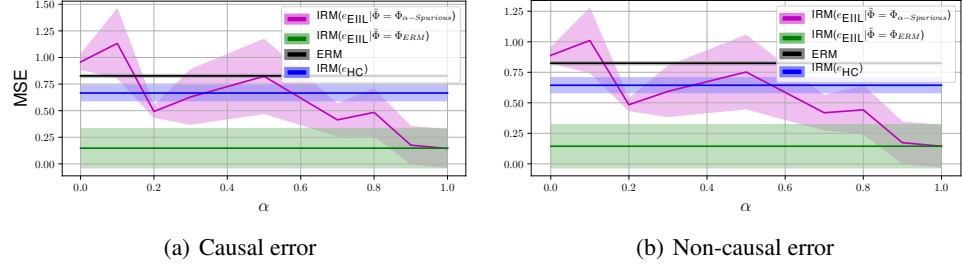

(a) Causal error        (b) Non-causal error

Figure 3: MSE of the causal feature $\mathbf{v}$ and non-causal feature $\mathbf{z}$. IRM($e_{\text{EIIL}}$)applied to the ERM solution (Black) out-performs IRM based on the hand-crafted environment (Green vs. Blue). To examine the inductive bias of the reference model $\tilde{\Phi}$, we hard code a model $\tilde{\Phi}_{\alpha-\text{SPURIOUS}}$ where $\alpha$ controls the degree of spurious feature representation in the reference classifier; IRM($e_{\text{EIIL}}$) out-performs IRM($e_{\text{HC}}$) when the reference $\tilde{\Phi}$ focuses on the spurious feature, e.g. with $\tilde{\Phi}$ as ERM or $\alpha$-SPURIOUS for high $\alpha$.

|  | Train accs | Test accs |
|---|---|---|
| ERM | $86.3 \pm 0.1$ | $13.8 \pm 0.6$ |
| IRM($e_{\text{HC}}$) | $71.1 \pm 0.8$ | $65.5 \pm 2.3$ |
| IRM($e_{\text{EIIL}}|\tilde{\Phi} = \Phi_{ERM}$) | $73.7 \pm 0.5$ | $68.4 \pm 2.7$ |
| IRM($e_{\text{EIIL}}|\tilde{\Phi} = \Phi_{Color}$) | $75.9 \pm 0.4$ | $68.0 \pm 1.2$ |
| $\Phi_{Color}$ | $85.0 \pm 0.1$ | $10.1 \pm 0.2$ |
| GRAYSCALE | $75.3 \pm 0.1$ | $72.6 \pm 0.6$ |
| ARL | $88.9 \pm 0.2$ | $20.7 \pm 0.9$ |

Table 6: Accuracy across ten runs with label noise $\theta_y = 0.25$ GRAYSCALE hard codes out the color feature and thus represents an oracle solution to CMNIST.

### E.2 COLORMNIST

Table 6 expands on the results from Table 1 by adding additional methods discussed in Section 3.

In Table 7 we measure the performance of some alternative strategies for optimizing the bi-level problem from Equation (EIIL). In particular, we consider alternating updates to the representation $\Phi$ and environment assignments $q$, as well as solving the inner/outer loop of EIIL multiple times. On the CMNIST dataset, none of these variants offers a performance benefit above the method used in Section 4.

EIIL$_{loops=k}$ indicates that the inner and outer loops of the EIIL objective in Equation (EIIL) are successively optimized $k$ times, with $k = 1$ corresponding to IRM($e_{\text{EIIL}}$), the method studied in the main experiments section. In other words, $\Phi_{loops=1}$ is solved using IRM($e_{\text{EIIL}}$), then this representation is used as a reference classifier to find $\Phi_{loops=k+1} = \text{IRM}(e_{EIIL}|\tilde{\Phi} = \Phi_{loops=k})$ in the

|  | Train accs | Test accs |
|---|---|---|
| EIIL$_{loops=1}$ | $73.7 \pm 0.4$ | $68.8 \pm 1.8$ |
| EIIL$_{loops=2}$ | $85.4 \pm 0.4$ | $10.0 \pm 0.4$ |
| EIIL$_{loops=3}$ | $75.6 \pm 0.5$ | $69.0 \pm 0.9$ |
| EIIL$_{loops=4}$ | $84.8 \pm 0.2$ | $10.1 \pm 0.3$ |
| EIIL$_{loops=5}$ | $76.2 \pm 0.5$ | $67.6 \pm 0.7$ |
| EIIL$_{AltUpdates}$ | $82.3 \pm 0.4$ | $24.0 \pm 1.1$ |

Table 7: We measure performance on CMNIST of various alternative approaches to optimizing the EIIL objective, ultimately concluding that none of the alternatives out-performs the method studied in Section 4. See text for details

|                            | Train accs     | Test accs      |
| -------------------------- | -------------- | -------------- |
| EIILv1                     | $68.7 \pm 1.7$ | $\mathbf{79.8 \pm 1.1}$ |
| EIILv1 (no regularizer)    | $78.6 \pm 2.0$ | $69.2 \pm 2.8$ |
| IRM (random environments)  | $94.7 \pm 0.1$ | $17.6 \pm 1.6$ |

Table 8: Our ablation study shows that both ingredients of EIILv1 (finding worst-case environments and regularizing invariance across them) are required to achieve good test-time performance on the ConfoundedAdult dataset.

next "loop" of computation. This also means that the training time needed is $k$ times the training time of IRM($e_{\text{EIIL}}$). As we expect from our theoretical analysis, using the IRM($e_{\text{EIIL}}$) solution as a reference classifier for another round of EIIL is detrimental: since the reference classifier already relies on the correct shape feature, environments that encourage invariance to this feature are found in the second round, so the EIIL$_{loops=2}$ classifier uses color rather than shape.

EIIL$_{AltUpdates}$ consists of optimizing Equation 1 using alternating steps to $\Phi$ and $q$. Unforatuntely, whereas this strategy works well for other bi-level optimization problems such as GANs, it seems to do poorly in this setting. This method outperforms ERM but does not exceed chance-level predictions on the test set.

### E.3 CENSUS DATA

**Ablation** Here we provide an ablation study extending the results from Section 4.2 to demonstrate that both ingredients in the EIILv1 solution—finding worst-case environment splits and regularizing using the IRMv1 penalty—are necessary to achieve good test-time performance on the ConfoundedAdult dataset.

From Lahoti et al. (2020) we see that ARL can perform favorably compared with DRO (Hashimoto et al., 2018) in adaptively computing how much each example should contribute to the overall loss, i.e. computing the per-example $\gamma_i$ in $C = \mathbb{E}_{x_i, y_i \sim p}[\gamma_i \ell(\Phi(x_i), y_i)]$. Because all per-environment risks in IRM are weighted equally (see Equation 2), and each per-environment risk comprises an average across per-example losses within the environment, each example contributes its loss to the overall objective in accordance with the size of its assigned environment. For example with two environments $e_1$ and $e_2$ of sizes $|e_1|$ and $|e_2|$, we implicitly have the per-example weights of $\gamma_i = \frac{1}{|e_1|}$ for $i \in e_1$ and $\gamma_i = \frac{1}{|e_2|}$ for $i \in e_2$, indicating that examples in the smaller environment count more towards the overall objective. Because EIILv1 is known to discover worst-case environments of unequal sizes, we measure the performance of EIILv1 using only this reweighting, without adding the gradient-norm penalty typically used in IRM (i.e. setting $\lambda = 0$). To determine the benefit of worst-case environment discovery, we also measure IRM with random assignment of environments. Table 8 shows the results, confirming that both ingredients are required to attain good performance using EIILv1.

## F ADDITIONAL THEORETICAL RESULTS

### F.1 OPTIMAL SOFT PARTITIONS MAXIMIMALLY VIOLATE THE INVARIANCE PRINCPLE

We want to show that finding environment assignments that maximize the violation of the softened version of the regularizer from Equation 3 also maximally violates the invariance princple. Because the invaraince principle $\mathbb{E}[Y|\Phi(X), e] = \mathbb{E}[Y|\Phi(X), e'] \forall e, e'$ is difficult to quantify for continuous $\Phi(X)$, we consider a binned version of the representation, with $b$ denoting the discrete index of the bin in representation space. Let $q_i \in [0, 1]$ denote the soft assignment of example $i$ to environment 1, and $1 - q_i$ denote its converse, the assignment of example $i$ to environment 2. Denote by $y_i \in \{0, 1\}$ the binary target for example $i$, and $\hat{y} \in [0, 1]$ as the model prediction on this example. Assume that $\ell$ represents a cross entropy or squared error loss so that $\nabla_w \ell(\hat{y}, y) = (\hat{y} - y)\Phi(x)$.

Consider the IRMv1 regularizer with soft assignment, expressed as

$$
\begin{aligned}
D(q) &= \sum_e ||\nabla_{w|w=1.0} \frac{1}{N_e} \sum_i q_i(e)\ell(w \circ \Phi(x_i), y_i)||^2 \\
&= \sum_e ||\frac{1}{N_e} \sum_i q_i(e)(\hat{y}_i - y_i)\Phi(x_i)||^2 \\
&= ||\frac{1}{\sum_i' q_i'} \sum_i q_i(\hat{y}_i - y_i)\Phi(x_i)||^2 + ||\frac{1}{\sum_i' 1 - q_i'} \sum_i (1 - q_i)(\hat{y}_i - y_i)\Phi(x_i)||^2 \\
&= ||\frac{\sum_i q_i \hat{y}_i \Phi(x_i)}{\sum_{i'} q_{i'}} - \frac{\sum_i q_i y_i \Phi(x_i)}{\sum_{i'} q_{i'}}||^2 + ||\frac{\sum_i (1 - q_i)\hat{y}_i \Phi(x_i)}{\sum_{i'} 1 - q_{i'}} - \frac{\sum_i (1 - q_i)y_i \Phi(x_i)}{\sum_{i'} 1 - q_{i'}}||^2.
\end{aligned}
\tag{4}
$$

Now consider that the space of $\Phi(X)$ is discretized into disjoint bins $b$ over its support, using $z_{i,b} \in \{0, 1\}$ to indicate whether example $i$ falls into bin $b$ according to its mapping $\Phi(x_i)$. Thus we have

$$
D(q) = \sum_b (||\frac{\sum_i z_{i,b} q_i \hat{y}_i \Phi(x_i)}{\sum_{i'} z_{i,b} q_{i'}} - \frac{\sum_i z_{i,b} q_i y_i \Phi(x_i)}{\sum_{i'} z_{i,b} q_{i'}}||^2 + ||\frac{\sum_i z_{i,b}(1 - q_i)\hat{y}_i \Phi(x_i)}{\sum_{i'} z_{i,b}(1 - q_{i'})} - \frac{\sum_i z_{i,b}(1 - q_i)y_i \Phi(x_i)}{\sum_{i'} z_{i,b}(1 - q_{i'})}||^2)
\tag{5}
$$

The important point is that within a bin, all examples have roughly the same $\Phi(x_i)$ value, and the same value for $\hat{y}_i$ as well. So denoting $K_b^{(1)} := \frac{\sum_i z_{i,b} q_i \hat{y}_i \Phi(x_i)}{\sum_{i'} z_{i,b} q_{i'}}$ and $K_b^{(2)} := \frac{\sum_i z_{i,b}(1 - q_i)\hat{y}_i \Phi(x_i)}{\sum_{i'} z_{i,b}(1 - q_{i'})}$ as the relevant constant within-bin summations, we have the following objective to be maximized by EIIL:

$$
D(q) = \sum_b (||K_b^{(1)} - \frac{\sum_i z_{i,b} q_i y_i \Phi(x_i)}{\sum_{i'} z_{i,b} q_{i'}}||^2 + ||K_b^{(2)} - \frac{\sum_i z_{i,b}(1 - q_i)y_i \Phi(x_i)}{\sum_{i'} z_{i,b}(1 - q_{i'})}||^2.
$$

One way to maximize this is to assign all $y_i = 1$ values to environment 1 ($q_i = 1$ for these examples) and all $y_i = 0$ to the other environment ($q_i = 0$). We can show this is maximized by considering all of the examples except the $i$-th one have been assigned this way, and then that the loss is maximized by assigneing the $i$-th example according to this rule.

Now we want to show that the same assignment maximially violates the invariance principle (showing that this soft EIIL solution provides maximal non-invariance). Intuitively within each bin the difference between $\mathbb{E}[y|e = 1]$ and $\mathbb{E}[y|e = 2]$ is maximized (within the bin) if one of these expected label distributions is 1 while the other is 0. This can be achieved by assigning all the $y_i = 1$ values to the first environment and the $y_i = 0$ values to the second.

Thus a global optimum for the relaxed version of EIIL (using the IRMv1 regularizer) also maximally violates the invariance principle.

