# OpenReview forum: "Exchanging Lessons Between Algorithmic Fairness and Domain Generalization"
_ICLR.cc/2021/Conference — Reject_

### Official Review · AnonReviewer2 · 2020-10-24
**ICLR 2021 Conference Paper3730 AnonReviewer2**

**Rating:** 4
**Confidence:** 3

**Review:**

Summary:
This paper studies the connections between algorithmic fairness and domain generalization. As discussed in Section 2, the “environment” in domain generalization plays a similar role as the “group membership” in algorithmic fairness. The paper shows in Table 2 that the methods of each field can apply to the other field.

The paper develops its own algorithm EIIL which extends the Invariant Risk Minimization (IRM) of domain generalization to work in the situation when the prior knowledge of environments is not available. And this extension is mainly based on the idea from algorithmic fairness literature which considers the worst-case environments and solves a bi-level optimization.

The paper shows empirically that their algorithm EIIL outperforms IRM with handcrafted environments in terms of test accuracy on CMNIST.

Strength:
(1) The connection between domain generalization and algorithmic fairness shown by the paper is interesting.
(2) The paper demonstrates the performance of EIIL via empirical results.

Weakness:
(1) Other than the high level intuitions and examples, the paper does not provide any theoretical analysis of the performance of the EIIL for domain generalization. What guarantees can EIIL get in terms of the test error and how does it compare to IRM (when making reasonable assumptions about the training and test distributions)?
(2) Similarly, the paper does not provide any theoretical analysis of EIIL for algorithmic fairness.
(3) On top of page 6, after explaining the bi-level optimization, the paper switches to the sequential approach (EIILv1) without much explanation. Why is the bi-level optimization not practical? How well can the proposed sequential approach approximate the bi-level optimization results and how does this affect the performance of EIILv1?

Reasons for score:
Overall I vote for rejection since the weakness outweighs the strength. The lack of theoretical analysis of the algorithm makes the paper incomplete.

Typo:
Page 5: two periods after word “poorly”.

---

> ### Author Response · Authors · 2020-11-24
> **Main points addressed above in rebuttal**
>
> Thanks for your helpful suggestions. In the main rebuttal we have addressed some of your suggestions regarding generalization properties of error, theoretical connections to fairness (through the invariance principle), and baselines.

---

### Official Review · AnonReviewer4 · 2020-10-27
**Interesting approach but lacks strong theoretical backing**

**Rating:** 5
**Confidence:** 3

**Review:**

### POST-REVISIONS ###
Thanks for the revisions made to the theoretical results. I still find parts of the discussion in Appendix F to be unclear.

Firstly, how do you derive eq. (5) from eq. (4)? In eq. (4), the denominators \sum_i q_i are independent of "\Phi(x_i)", but in eq. (5), they have a dependence on \Phi through z_{i,b}. I think the change in normalization important to show the invariance principle holds (as the invariance principle requires a conditioning on each value \Phi takes), but am unable to follow your derivation.

Secondly, I'm not convinced that the maximizing partition for eq (5) assigns all examples with y=1 to one group, and those with y=0 to another group. Wouldn't the maximizing partition also depend on what \hat{y} evaluates to for those examples?

Overall, I'm able to see what the authors are trying to get at with this example, but unfortunately the revisions aren't sufficient to address all of my concerns regarding the theoretical results.
********************

The paper presents an approach for training models that generalize well to out-of-distribution (OOD) samples, particularly when the source of domain shift (e.g. spurious correlations or sensitive groups) is not known before hand. The paper combines ideas from two prior papers from the domain generalization and fairness literature: (i) invariant  risk minimization for OOD generalization (Arjovsky et al.) and (ii) adversarially reweighting for fairness without protected groups (Lahoti et al.).
At a high level, the proposed approach seeks to minimizes the average classification loss across the worst-case partitioning of the dataset into two groups. Experimental results on datasets with synthetically generated spurious features show that the proposed approach is able to generalize better to OOD samples in the high noise regime, without having knowing aprior which features are spuriously correlated with the labels.

Pros:
- The question tackled is practically important: how one can generalize to OOD samples without knowing the exact source of discrepancy between train and test data.
- The experimental results look encouraging

Cons:
- The paper lacks a clear theoretical motivation for the specific optimization objective that the authors end up using (eq 3). In particular, do we know (at least in some a simple setting) that maximizing this objective over soft-group memberships "u_i" will identify the partitioning of the data that maximally violates the Invariant Constraint? I elaborate on this next.

Relaxed training objective lacks strong theoretical backing:
The authors directly adapt the training setup of Arjovsky et al., where the goal is to train a model which learns the same conditional label distribution for any given input "x" across a set of known partitioning of the training data, dubbed as the invariance constraint . Each of these partitions, referred to as 'environments', represent a different training distribution, and the goal is to train a model that performs equally well across all of them. Arjovsky et al. show that for the special case of linear invariant predictors, the training problem can be relaxed into an unconstrained objective with a regularization penalty.

The present paper extends the setup of Arjovsky et al. to problems where the environments are not a prior known, and seeks to minimize the average classification loss over a partitioning of the data that maximally violates the invariance constraint. However, they do not explicitly solve this optimization problem, and instead simply minimize the worst-case value of the "relaxed training objective" of Arjovsky et al. over all (soft) partitioning of the data.

Is the relaxation that Arjovsky et al. employ with known environments still relevant to your problem formulation, where you would like the invariance constraint to hold for all possible partitioning of the data?

At the very least, this requires a discussion. Ideally, it would be nice to see a derivation of the relaxation for some simple special cases: e.g. like Arjovsky et al., can you show that for linear predictors, "finding a partition that maximally violates the invariant constraint" is equivalent to "maximizing the relaxed unconstrained objective in eq. 3 over partitions"?

Other comments:
- Eq 3:  I think "w" is a scalar here (otherwise evaluating the gradient at w = 1.0 doesn't make sense). Please make that explicit and also provide some intuition for why this regularization penalty with a scalar "w" makes sense for your problem set up.
- I am not entirely sold on the general theme of this paper of exchanging lessons between fairness and domain generalization. The authors are definitely correct in crediting a prior fairness paper for the idea of  the idea of adversarially re-weighting examples with a soft groups model, but as they themselves point out this idea has existed in different forms in the domain generalization literature (e.g. DRO). So my reading is that the paper seems to slightly over-emphasize the connection to the fairness literature, but this is a personal take. Having said this, the paper does provide (in Sec 2) a nice literature overview of similar problems tackled by the domain generalization and fairness communities.
- In the color MNIST experiments, you observe "IRM(eEIIL) generalizes better than IRM(eHC) with sufficiently high label noise". If I understand correctly, IRM(eHC) has access to the true environments, whereas IRM(eEIIL) uses environments inferred from data. Wouldn't we expect the former method to have an advantage over the latter?
- Additional baseline: Would it make sense to compare with (a form of) DRO for the color MNIST task (e.g. ones cited in Table 2)?  You do mention in another experiment that Lahoti et al. compare with DRO for their particular fairness application, but do those observations also apply yo the tasks you consider in this paper.
- Iterative training: I think a natural extension of your approach (which you've probably already thought about) is to solve (EIIL) using an iterative technique that alternates between maximizing over "q" and e.g. performing gradient descent updates on "\Phi". Iteratively performing full optimizations over both sets of parameters may not in general have good convergence properties.
- Might be a relevant citation for the use of soft partition assignments for fairness: https://arxiv.org/pdf/2002.09343.pdf
- Fig 1: Would be nice if the plots were color blind friendly :)
- References: Might be good to mention the conference venue wherever available: e.g. Hashimoto et al. appeared in ICML 2018.

---

> ### Author Response · Authors · 2020-11-24
> **Following up on particulars**
>
> Thanks for your helpful suggestions. Beyond the main points of our rebuttal above, here are several specific responses to your review
> * We have updated the reference to include conference names where appropriate.
> * About the plot colors, we used the seaborn “colorblind” palette for the original submission, so we hope that it is already relatively colorblind friendly. We remain open to suggestions about how to further improve on accessibility of the plots.
> * Thanks for pointing us to the soft partitions paper, which we now cite.
> * We use the w=1.0 notation in the same way as Arjovsky et al. For multi-class classification it is equivalent to multiplying all dimensions of the representation by 1.0 (i.e. uniform diagonal loading) prior to the softmax.
> * To your question about under what conditions solving the objective from eqn 3 satisfies the invariance principle, see the fourth bullet in our main response to all the authors.

---

### Official Review · AnonReviewer3 · 2020-10-30
**Interesting connection but could be supported with more theoretical guarantees**

**Rating:** 6
**Confidence:** 2

**Review:**

The main contribution of the paper is to highlight the similarity between two active areas in ML namely "domain generalization" and "fairness". Further, the paper proposes an approach inspired by recent developments in the fairness literature for domain generalization. The high-level idea is that similarly to the way that fair algorithm are able to improve the worst-case accuracy of predictors across different groups without knowing the sensitive attributes, perhaps we can use these ideas to domain generalization when environment partitions are not known to the algorithm. In some sense, in both of these research areas the goal is to design robust algorithms. Similarly, the paper uses the idea from domain generalization to design fair algorithms w.r.t. a notion called "group sufficiency". The idea is to somehow infer the "worst-case" subgroup (i.e., the one that our algorithm has the worst accuracy on it) and then using a round of auditing improve the performance of the algorithm across all subgroups.

The authors have supported their approach with empirical evaluations. In particular, I find the result on CMNIST quite interesting where the new algorithm as opposed to the standard approach like ERM will not be fooled by the spurious feature and can infer the useful environment.

While the paper has introduced (to best of my knowledge) a new concept, it seems that are many interesting questions that could show the applicability of the connection better are not yet answered (e.g., bi-level optimization EIIL). This could also help the paper to be supported with more provable guarantees. In general the paper is exploring a new connection between two areas and has shown its efficacy in practice and I believe it can lead to further works on this topic.


Minor comments:
- define the notion of "group sufficiency" explicitly in the paper. I could not find the definition of the notion in words till in the caption of Figure 2 on page 8 and is formally defined on page 12!
-page 5: poorly. . Consider -> poorly. Consider
-page 6: generalizattion -> generalization
-page 7: graysacle ->grayscale
-page 14: exagerated -> exaggerated
-page 14: orginal -> original
-page 17: implicily -> implicitly

---

> ### Author Response · Authors · 2020-11-24
> **Typos fixed**
>
> Thanks for your time in reviewing our work, and for your helpful feedback. We fixed the typos you pointed out in the revision.

---

### Official Review · AnonReviewer5 · 2020-11-07
**Insightful paper but weak results**

**Rating:** 4
**Confidence:** 4

**Review:**

This paper presents parallels between algorithmic fairness and domain generalization literatures. The authors explore a learning setup where the goal is to learn some representation $\Phi(x)$ that is "independent" of some environmental variable $e$. The authors explore cases where $e$ is known or not and come up with some algorithms that draw connections between recent work on domain generalization, specifically invariant risk minimization (Arjovsky et al 2019) and fairness. The authors conclude the paper with three different examples addressing domain generalization and fairness. While the supporting experimental results are not very strong, the connections observed are interesting. I have several points for clarification that I detail below.

**Exposition.** While the introduction of the paper is written nicely and the ideas are communicated nicely, the mathematical exposition and presentation in this paper is not self-contained. For example, it is close to impossible for a reader to follow this paper without having read (Arjovsky et al 2019) and (Liu et al. 2018). I suggest that the authors expand on the mathematical exposition, defining all terms, and explaining different things came from, especially since the paper is supposed to have a broad audience across two communities.

**Convergence of Algorithm.** The proposed algorithm comes with no guarantees and I suspect it will not converge in a variety of situations, especially in cases where $C^{\text{IRM}}$ is nonconvex in (EIIL). Why would you need to run the inner optimization and outer optimizations to convergence each time? Have you tried a GDA version of the algorithm?

**Cost of the Algorithm.** The proposed algorithm is costly because IRM has to be solved multiple times before it converges. Can you please comment on the computational cost of the proposed algorithm as compared to ERM and other baselines in each experimental setup?

**Limitations of generalization-first fairness.** This section is nicely written and much appreciated.

**Choice of $\Phi_{spurious}$.** The algorithm is sensitive to initialization choice of $\phi_{spurious}$  as the authors also find with their Color MNIST experiments. In particular, there is a huge performance gap in Fig 1.b. for $\theta_y \in (0, 0.15)$ that needs to be addressed. On the other hand, the algorithm seems to work well in the severely overfitting regime where ERM can be thought of having learnt $\Phi_{spurious},$ as also discussed by the authors in the second paragraph of Section 3.2. However, the real world is not so black and white and hence this poses a severe limitation. Can you please explain?

**Connection with (Liu et al. 2018).** In the third paragraph of **Fairness** section in Page 4, the authors claim a connection between IRMv1 (btw, IRMv1 is misspelled as IMRv1 there) and (Liu et al. 2018). This connection is not obvious to me. Can the authors make it rigorous?

**Continuous $e$.** Can you please comment on how this setup may generalize to continuous $e$? At least can you please comment on the scaling of the algorithm with the cardinality of the set of environments?

**Theorem 1.** Unfortunately, Theorem 1 and entire Section 3.2 is only applicable to a severely unusual and overfitting case (similar to the Color MNIST example) where there is perfect correlation between the environment variable and the label. The fact that the algorithm works well in this situation is not surprising. The real world, however, is not black and white and the limitations of the proposed framework in real-world situations (similar to $\theta_y \in (0, 0.15)$ in Fig 1.b.) remain to be understood.

**Confounded Adult Dataset.** Please make it clear that this is constructed by the authors. I only understood that when started looking for the details in Appendix.

**Typos.** (1) In Eq. (2), $w\circ\Phi$ should be replaced with $w.\Phi$. (2). second paragraph of **Fairness** paragraph on Page 4, the word attribute is missing after "sensitive".

Overall, while the subject area of the paper is exciting, unfortunately, the execution (both empirical and theoretical) is weak. I tend to remain to vote for rejection with encouragement for a more thorough empirical and theoretical investigation of the problem.

---post rebuttal---

After reading the authors' response, the other reviews, and the revision to the paper, I find that my comments are not sufficiently addressed. The author did not even acknowledge the existence of the prior work, REPAIR, in the revised paper. The imprecise mathematical expressions are still in the paper despite feedback from multiple reviewers. From a practical point of view, the developed algorithm is not scalable as it requires to (almost) solve the inner maximization at each iteration (based on the rebuttal), and it only works in the significantly overfitting regime (the authors are yet to show its performance in a more interesting regime). From a theoretical point of view, the applicability of the theory is also extremely limited to the perfectly overfitting regime, which does not capture the real world. In addition, I agree with AnonReviewer4 that the proofs are inscrutable.  I regret to say that despite the fact that the subject area of the paper is exciting, I am adjusting my score to 4 post rebuttal.

---

> ### Comment · AnonReviewer5 · 2020-11-12
> **Missing key reference**
>
> One more comment that I want to append to my original review:
> The idea of making connections between algorithmic fairness and domain generalization has been explored in the literature as early as REPAIR (https://arxiv.org/abs/1904.07911), which is missing as a reference in this paper. This will discount my favorable impression for making that connection, based on which I am lowering my score from 6 to 5. REPAIR also does experiments on Color MNIST (which is the main experimental setup in this paper).
>
> Li, Y. and Vasconcelos, N., 2019. REPAIR: Removing representation bias by dataset resampling. In Proceedings of the IEEE Conference on Computer Vision and Pattern Recognition (pp. 9572-9581).

---

> > ### Author Response · Authors · 2020-11-23
> > **REPAIR does not discuss the connections between domain generalization and algorithmic fairness**
> >
> > Thanks very much for your detailed review. We are in process of revising the paper and submitting a more comprehensive rebuttal to address some of the concerns you brought up in the main review (which are helping us to make the paper stronger). In the meantime we wanted to briefly address this extra suggestion about the REPAIR paper. Thank you for pointing us to this work on example reweighting for addressing dataset bias, which we will cite in the revision. However we disagree with your assessment that there is overlap between this paper’s contributions to the literature and ours. The REPAIR method adaptively reweights the per-example contributions to the overall risk, and is not a domain generalization paper as you suggested (there is no notion of training under multiple domains/environment with the hopes of generalizing well to a held-out domain). It is better understood as a robust optimization paper. Also, the connection to algorithmic fairness offered by the REPAIR paper is rather cursory. For example, they do not establish the connection between example reweighting in their method and example reweighting in fairness methods that deploy distributionally robust optimization (e.g. Hashimoto ICML 2018).
> >
> > Thanks again for your time in reviewing our work, and we hope the revision (to be posted shortly) will address the remainder of your suggestions.

---

> ### Author Response · Authors · 2020-11-24
> **Following up on a few points**
>
> Thanks again for your time in reviewing our paper. Here are some responses to minor concerns from your original review:
> * The connection from IRM to Liu et al is through the invariance principle. Liu et al focus on the group-sufficiency principle from the fairness literature, which as we point out in our paper, is equivalent to the invariance principle in domain generalization, discussed in the IRM paper. The regularizer in IRMv1 was previously proposed as a way of optimizing the invariance principle. We have included in the appendix of the revision a proof showing a condition under which optimizing our proposed softened version of the IRMv1 regularizer will optimize the invariance principle.
> * The cost of EIILv1 is roughly triple that of IRM or ERM, since we need to first solve ERM for a reference classifier, then solve the inner loop of EIIL (there tend to be fewer per-example weights than network parameters in the settings we describe so this goes quickly), then finally solve IRM with the inferred environments. So we can think of implementing EIIL as roughly the training cost of implementing an ensemble of three networks (but same memory/inference cost as a single network), noting that ensembles tend to fail for the sort of dramatic test-time shifts that we study.
> * We take care to clarify in the experiments that we have constructed the ConfoundedAdult dataset in order to measure out-of-distribution performance
> * The domain generalization literature tends to work with discrete rather than continuous domains; because the general strategy of EIIL is to find worst-case environments w.r.t. a specific DG learning algorithm, the application to continuous domains is not obvious. In terms of scaling with increasing cardinality of a discrete domain set, the theoretical properties of IRM suggest that the more (statistically independent) environments the better in term of generalization guarantees. This suggests that extending EIIL to find more than two environments (with a term to promote diversity amongst inferred environments) may further help out-of-domain generalization. While this direction is left for future work, it is now discussed as a footnote in the revision.

---

### Author Response · Authors · 2020-11-24
**Rebuttal**

We would like to thank the reviewers for their detailed and extremely helpful reviews. We have updated the manuscript to incorporate these suggestions, Below we summarize how we addressed the common concerns here, and will follow up on a per-reviewer basis to address the remaining issues.
* Some reviewers were concerned that we only measured a win when evaluating EIIL in the high label noise regime or for certain choices of reference classifier. This highlights an important aspect of our paper -- our focus is on situations in which ERM performs poorly. This captures a wide range of problems in machine learning (see the Shortcuts paper [2]). We study the high label noise not because it is compelling on its own right, but rather because it serves as a controllable proxy for these cases, in which the ERM reference classifier is sub-optimal. Note that our interest in failure modes of ERM makes it challenging  to derive formal guarantees about EIIL without introducing some assumptions over the ERM behavior (this is why we make such assumptions in our theorem).
* One reviewer suggested that hand-crafted environments should out-perform inferred worst-case environments, but an important point of our work is that this is not necessarily the case. As we mention in Section 3 (and show empirically in Section 4 and theoretically in Appendix B.2), even when hand-crafted environments are available they can sometimes be improved upon by EIIL. Even when environments/domains are known, they may be suboptimal from the perspective of learning an invariant representation. EIIL tends to find more dramatically different environments, which in turn helps IRM find a good global optimum by making the learning signal through the regularization term more informative.
* As the reviewers have pointed out, satisfying the invariance principle is the most important objective to establish a connection to fairness. But it leaves open the question of whether the specific regularizer used in IRMv1 is the best way to achieve the invariance principle (this is an open question in general for domain generalization). We provide new theoretical results in Appendix F showing that maximizing the soft/relaxed version of the IRMv1 regularizer using inferred environments (which is the goal of EIILv1) also maximally violates the invariance principle.
* In terms of theoretical guarantees related to generalization on held-out domains, we inherit all the generalization properties of IRM so long as the EIIL solution remains in the same degree of ‘linear general position’ (LGP) as the hand-crafted environments. When domains are Gaussian distributed, the LGP degree can be thought of as the inherent rank of the union of training domains mean vectors (the recent ‘Risks of IRM’ paper [1] does a good job of clarifying this in their appendix). So the EIIL solution can be expected to maintain the LGP degree of the hand crafted domains so long as its partitions do not induce two statistically identical environments. Anecdotally we can say that this does not happen in practice (the inferred environments tend to be quite distinct). Formally, we can appeal to theorem 10 of the IRM paper, stating that the set of covariance matrices that do not lie in LGP (assuming environments come from the linear SEM) is measure zero. Noting that covariances matrices under the EIIL-discovered environments q(X|e) can be expressed in terms of expected covariances under the SEM distributions p(X|e) multiplied by an importance weight q(X|e)/p(X|e), then so long as q(X|e)/p(X|e) is nonzero, the covariances remain positive semidefinite, theorem 10 still holds, and we have the same generalization properties as IRM.
* We add three new baselines for CMNIST experiment in Appendix E.2) following reviewer suggestions. ARL performs better than ERM but still worse than random chance on the test distribution (this is not surprising since distributionally robust methods like ARL are well-suited for smaller test time distribution shifts, not the more drastic intervention in CMNIST that reverses all color correlations). We also add alternating updates between the IRM and EIIL steps (i.e. alternating a single gradient step on the inferred environments update with a single gradient step on the representation update). Unfortunately this strategy, which tends to work well in other bi-level problems like GANs, does not seem to work effectively in our initial studies, again outperforming ERM but achieving below chance rates on the test set. Finally we try optimizing the inner and outer loop multiple times. This strategy induces an oscillation between the correct shape-based classifier and the incorrect color-based classifier, again highlighting the relevance of the reference classifier used by EIIL

[1] Roesnfeld et al, The Risks of Invariant Risk Minimization, preprint, https://arxiv.org/abs/2010.05761

[2] Geirhos et al, Shortcut Learning in Deep Neural Networks, https://arxiv.org/abs/2004.07780

---

### Decision · Program_Chairs · 2021-01-07
**Final Decision**

**Decision:**

Reject

**Comment:**

The paper analyzes connections between algorithmic fairness and domain generalization literatures. The reviewers found the paper interesting but they also raised some important concerns about it.

The applicability of the method presented in the paper is not clear nor well-discussed in the paper.

The papers and the revised version do not not cite important related work.

The mathematical exposition in the paper is a bit hard to read. Even after revision, the reviewers find part of the paper(Appendix F) very hard to read.

Overall, the paper in the current version is below the high acceptance bar of ICLR.